# Impacts of Social Participation on Self-Rated Health of Aging Women in China: With a Mediating Role of Caring for Grandchildren

**DOI:** 10.3390/ijerph18115790

**Published:** 2021-05-28

**Authors:** Shuliu Tian, Lei Xu, Xiangling Wu

**Affiliations:** School of Political Science and Public Administration, Wuhan University, Wuhan 430072, China; shuliutian@whu.edu.cn (S.T.); xulei19891014@163.com (L.X.)

**Keywords:** social participation, self-rated health, caring for grandchildren, aging women, CHARLS

## Abstract

Population aging is a global challenge and the degree of population aging is continuing to deepen in China. Under the active aging policy framework by WHO, great importance has been attached to aging women and participation is emphasized for the well-being of the elderly. This study aimed to investigate the relation between social participation and self-rated health status of aging women in China and whether caring for grandchildren mediated such an association. Adopting data from the 2018 China Health and Retirement Longitudinal Study (CHARLS), this study used Oprobit regression, propensity score matching (PSM), and instrument variable regression to estimate the effects. The result showed that there was a positive association between social participation and self-rated health among aging women in China, and social activities that directly made contributions to others had the most significant impacts on self-rated health. Furthermore, the mediator analysis confirmed that caring for grandchildren played a role between social participation and self-rated health. In conclusion, to deal with population aging challenges, the society should recognize the value of intergenerational care for aging women and the government need to strengthen policy supports to guarantee platforms and opportunities for the elderly to participate in social activities.

## 1. Introduction

According to the United Nations (UN), a country enters into an aging society when the population of people over 60 years old accounts for more than 10% of the total population or the population of people over 65 years old accounts for more than 7% of the total population [1]. In 1999, China officially announced that it had entered an aging society. Since the 21st century, the rate of population aging has been accelerating and the degree of population aging has been continuing to deepen in China. According to data released by the National Bureau of Statistics of the People’s Republic of China, by the end of 2019, the number of people aged 60 and over in China has reached 254 million, accounting for 18.1% of the total population; and the number of people aged 65 and over has reached 176 million, accounting for 12.6% of the total population [2]. The China Development Research Foundation (CDRF) published China Development Report 2020: Development Trends and Policies of China’s Population Aging in June 2020. The report predicted that China’s population over the age of 65 would make up 14% of its total population by around 2022 and that China would formally transit to an aging society by then. The report also foresaw that population aging in China would reach its peak in 2050 and China’s population over the age of 65 would make up 27.9% of its total population.

Nevertheless, population aging is not unique to China. It is a global population development trend and a governance problem faced by many countries in the world. The World Health Organization (WHO) initiated that governments should enact active aging polices to deal with challenges of population aging and emphasized benefits of participation for the elderly. According to the WHO, one of the major challenges of aging population is the feminization of aging, regarding the fact that women live longer than men almost everywhere [3] (p. 39). However, women’s traditional roles as family caregivers may also contribute to their increased poverty and ill health in older age [3] (p. 20). As a result, the health problems of older women are more serious, and improving the health of aging women has become a widespread concern in the society. Hence, we ask, how is the social participation of aging women in China? Does social participation affect their health? To what extent is the impact of health among aging women? These issues are worthy of study. In this article, we chose self-rated health as the representative of an individual’s health status and took a deeper look at the relations between social participation and self-rated health of aging women in China and the mediating role of caring for grandchildren in this relationship. We started the analysis with a comprehensive literature review. In order to better construct our hypotheses for the research, we summarized the development of active aging theory and presented existing results on social participation, self-rated health, and intergenerational care.

## 2. Literature Review

Under the global phenomenon of population aging, academic researchers have paid close attention to the theory and practice of aging issues. Research on aging has undergone multiple stages of development and evolution and has achieved a transition from a passive aging attitude to an active aging attitude. Early in the last century, traditional cultural values, economic environment, and institutionalized life led to the emergence of a dark age of aging, namely, the era of negative aging [4]. The concept of successful aging popularized in the 1980s promoted the transition from negative aging to positive aging [5]. Furthermore, the concept of productive aging by Robert Butler began to focus on the social participation of the older groups [6]. Later, the concept of active aging, inspired by the activity theory of aging of Robert J. Havighurst, extended the social participation of the aging population from the economic field to all aspects of the society [7,8]. In 2002, the WHO published the report Active Aging: A Policy Framework and stated that “active aging is the process of optimizing opportunities for health, participation and security to enhance quality of life as people age” [3] (p. 12). This framework treated participation as a basic human right and defined it as full participation in socioeconomic, cultural, and spiritual activities whether it is paid or unpaid [3] (p. 46). In China, the Chinese government issued the Mid-term and Long-term Plan for the State to Actively Respond to Population Aging in November 2019, and in October 2020, the Government upgraded the active aging policy as a national strategy.

Concerning benefits of social participation to older people, Cottrell et al. used role theory to explain the importance of active social participation for senior citizens to adapt to their retired lives from a sociological perspective [9]. Hooyman and Kiyak believed that voluntary organizations are an important mechanism to promote the reintegration of older people into the society [10]. Shi concluded that the higher degree of social participation of the elderly, the more social and intergenerational support they can obtain, and the lower risk of being abused by family members [11]. Additionally, social participation has a positive impact on physical health among older people [12,13,14]. In addition, many scholars have found that the active participation of the elderly is beneficial to their subjective well-being or mental health, specifically, to lower depression [12,13,14], improve functional and self-rated health [15], enhance happiness, and realize the value of life [16,17].

As for functions of self-rated health, it is a common health indicator frequently employed in sociological health research since the 1950s [18]. Self-rated health is based on interviewees’ evaluations of their own health and it is a subjective measurement that depends on one’s complex perceptions and hypothesis [18,19,20,21]. Many outstanding scholars have clarified the association of self-rated health with its predictive power on mortality [18,22], Additionally, a large number of studies have shown that self-rated health is a reasonable representative of an individual’s health status [17,18,19,20,21,22,23].

Another highlight of our study is the choice of the mediator. After the implementation of the one-child policy in the late 1970s in China, family structure has shown a trend of miniaturization. The “4-2-1” family structure, which includes four elder people (grandparents-in-law and grandparents), two young people (parents), and a teenager (the only child) has become common in China, while small average household sizes, of fewer than three persons per household, were found in most countries of Europe and Northern America [24]. In most European countries, it is unusual for older people to live with their adult children, the family size becomes smaller, and fewer than 30% of the households include children [24]. Chinese people over 40 years old are now responsible for both supporting their families and caring for the elderly. However, due to heavy workloads and a fast-paced modern life, they often have no time to take care of the next generation. This situation has created an intergenerational gap in parenting and leaving the kids with their parents has become a popular choice for young parents [25]. Under the complicated contexts of rapid aging population, fast social transformation, and speedy urbanization, the phenomenon of intergenerational care has become a common social trend in China [26] and grandparent caregiving has become more common for the Chinese elderly [27,28,29]. Furthermore, research has proved that grandmothers play a more important role in the care of grandchildren [30,31,32,33,34]. Therefore, there might be some connections among social participation, self-rated health, and intergenerational care of aging women in China.

While existing research focuses on the general aging group and have testified the benefits of social participation, especially the impact on health-related factors, few studies have investigated the effects of this variable in a population of older women. Specifically, few studies consider the mediating impact of grandmother caregiving responsibility between their health status and social participation. Given this, we attempted to examine the relationship between social participation and self-rated health among Chinese aging women, with the mediating effect of caring for grandchildren taken into account. The conceptual framework is shown in Figure 1.

Specifically, we aimed to investigate the following two research questions and associated hypotheses:

**Question** **1** **(Q1).***Does social participation have any impact on the self-rated health among aging women in China*?

**Hypothesis** **1** **(H1).**
*We hypothesized that there was a positive association between social participation and the probability of a better self-rated health among aging women in China (Path A).*


**Hypothesis** **1a** **(H1a).**
*Under the circumstances that H1 was valid, we then hypothesized that different types of social participation might have different effects on the probability of better self-rated health among aging women in China.*


**Question** **2** **(Q2).***How does caring for grandchildren mediate such a relationship*?

**Hypothesis** **2** **(H2).**
*We hypothesized that caring for grandchildren might play as a mediating role between social participation and the probability of a better self-rated health among aging women in China (Path B1 and Path B2).*


## 3. Material and Method

### 3.1. Data

This article used data from the China Health and Retirement Longitudinal Study (CHARLS) 2018. CHARLS is a large-scale interdisciplinary survey project hosted by the Institute of Social Science Survey of Peking University. It is a longitudinal survey aiming to collect a high quality nationally representative sample of Chinese residents of ages 45 and older to serve the needs of scientific research on the elderly. In order to ensure sample representativeness, the CHARLS baseline survey covered 150 countries/districts and 450 villages/urban communities across the country, involving 17,708 individuals in 10,257 households, reflecting the middle-aged and older Chinese population collectively [35]. This article defined aging women based on the international standard, namely, women aged 60 and above. After screening and excluding invalid samples, 5015 valid samples were obtained.

### 3.2. Variables Selection

#### 3.2.1. Outcome Variable

The outcome variable of this research was self-rated health. In the CHARLS 2018 questionnaire, the variable was measured by the questions: Would you say your health is very good, good, fair, poor, or very poor? Answers in this research were recorded from 1 (very poor) to 5 (very good).

#### 3.2.2. Explanatory Variables

The explanatory variables of this research were social participation, specifically, whether the individual took part in any social activities. In the CHARLS 2018 questionnaire, social participation was measured by the question: Have you done any of these activities in the last month? This question had 12 choices: (1) interacted with friends; (2) played mahjong, played chess, played cards, or went to a community club; (3) provided help to family, friends, or neighbors who do not live with you; (4) went to a sport club, social club, or other kind of club; (5) took part in a community-related organization; (6) did voluntary or charity work; (7) cared for a sick or disabled adult who does not live with you; (8) attended an educational or training course; (9) made an investment in stock; (10) used the internet; (11) other, or (12) none of these. If any of the first 11 choices were selected, older women were considered to have participated in social activities, otherwise, they were considered to not have participated in social activities. Individuals who responded “(12) none of these” were recorded as “0 = no”, and all other choices were recorded as “1 = yes”.

#### 3.2.3. Mediator

The mediator we chose was caring for grandchildren. Considering the limited choices in the questionnaire, it has been proved that women spend more time as family caregivers and express a greater sense of responsibility towards family members [36,37,38]. In the CHARLS 2018 questionnaire, caring for grandchildren was measured by the question: During last year, did you/your spouse spend time taking care of your grandchildren? This question had 3 choices: (1) yes; (2) no; or (3) I have no grandchild. Interviewees who responded yes were recorded as “1 = yes”; others were recorded as “0 = no”.

#### 3.2.4. Control Variables

We selected control variables based on previous studies about factors that affect the health status of older women and included the following variables: age, residence, education, smoking, drinking, education, marriage, minority, income, insurance, and pension [39,40,41,42,43]. Table 1 presents the control variables and the original questions from the CHARLS 2018 questionnaire and codes in this research.

#### 3.2.5. Instrumental Variable

Effective instrumental variables (IV) need to meet two requirements at the same time: (1) the selected IV and the endogenous explanatory variable (social participation) must have a correlation; (2) the selected IV must be irrelevant to the outcome variable (self-rated health) [44]. Based on the requirements above, we selected the relationship with children as the IV for our research. In the CHARLS 2018 questionnaire, the relationship with children was measured by the question: How satisfied are you with your relationship with children? This question had six choices: (1) completely satisfied; (2) very satisfied; (3) somewhat satisfied; (4) not very satisfied; (5) not at all satisfied; or (6) no children now. The answers in this research were recorded from 1 (not at all satisfied) to 5 (completely satisfied), while answer “(6) no children now” was excluded as it had a very small number of samples.

### 3.3. Core Models Development

#### 3.3.1. Oprobit Model

The outcome variable “Self-Rated Health” was recorded from “1–5” as an ordered categorical variable. Accordingly, we chose the Oprobit regression model for this research and used latent variables to calculate maximum likelihood estimation (MLE).

The specific model is as follows:P(SH = SH_i_|X,β) = P(SH = SH_1_|x_1_,x_2_,x_3_…,x_k_,)(1)

In model (1), i refers to the individuals in the survey; SH refers to the self-rated health of the individual i. In the Oprobit model, we introduced a latent variable ${\mathrm{SH}}_{i}^{*}$, which cannot be directly observed as the self-rated health. ${\mathrm{SH}}_{i}^{*}$ meets the following form:
(2)$${\mathrm{SH}}_{i}^{*}=\beta_{1}SP_{i}+{ϒ}_{1}Z_{i}+\varepsilon_{i},\;\varepsilon_{i}∼\mathrm{Normal}(0,1)$$

In model (2), i refers to the individuals in the survey; ${\mathrm{SH}}_{i}^{*}$ refers to the latent variable of self-rated health of the individual i; SP_i_ refers to the social participation of the individual i; and Z refers to other control variables, including age, residence, smoking, drinking, education, marriage, minority, insurance, income, and region. β_1_ is the parameter estimate; ϒ_1_ is the vector of parameter estimate; ε_i_ is the error term.

#### 3.3.2. Mediating Effect

To test Hypothesis 2, we constructed recursive formulas from model (3) to model (5), and model (4) was the same as model (2).
(3)$${\mathrm{SH}}_{i}^{*}=\alpha_{0}SP_{i}+\lambda_{0}Z_{i}+\delta_{i}$$
CFC_i_ = α_1_SP_i_ + λ_1_Z_i_ + τ_i_(4)
(5)$${\mathrm{SH}}_{i}^{*}=\alpha_{2}SP_{i}+\alpha_{3}\mathrm{CF}C_{i}+\lambda_{2}Z_{i}+\Psi_{i}$$
where i refers to the individuals in the survey; ${\mathrm{SH}}_{i}^{*}$ refers to the latent variable of self-rated health of the individual i; SP_i_ refers to the social participation of the individual i; CFC_i_ is the mediator, representing the responsibility of caring for grandchildren of individual i; and Z_i_ is control variables, including age, residence, smoking, drinking, education, marriage, minority, insurance, income, and region. α_0_, α_1_, α_2_, α_3_ are parameter estimates; λ_0_, λ_1_, λ_2_ are the vectors of parameter estimates; δ_i_, τ_i_, Ψ_i_ are the error terms in corresponding models.

#### 3.3.3. Propensity Score Matching (PSM)

According to the PSM method, we supposed y_i_ as the outcome variable of self-rated health of aging women,
$y_{i}^{1}$ as the self-rated health of aging women with social participation, and
$y_{i}^{0}$ as the self-rated health of aging women with social participation. The experimental group (with social participation) and the control group (without social participation) were matched separately, and the differences between the two groups were compared. We then calculated the average treatment effect for the treated (ATT), which is defined as:
(6)$$\mathrm{ATT}=E(y_{i}^{1}|X_{i}=1)-E(y_{i}^{0}|X_{i}=1)$$

#### 3.3.4. Instrumental Variable Regression

In this paper, the instrumental variable was “Relationship with Children”, and we used two-stage least squares (2SLS) and limited information maximum likelihood (LIML) to conduct endogeneity testing. The specific model is as follows:
(7)$${\mathrm{SH}}_{i}^{*}={ώ}_{1}SP_{i}+{ώ}_{2}IV_{i}+\kappa Z_{i}+\rho_{i}$$
where i refers to the individuals in the survey; ${\mathrm{SH}}_{i}^{*}$ refers to the latent variable of self-rated health of the individual i; SP_i_ refers to the social participation of the individual i; IV is the instrumental variable, representing the relationship with children of individual i; and Z_i_ is control variables, including age, residence, smoking, drinking, education, marriage, minority, insurance, income, and region. ώ_1_ and ώ_2_ are parameter estimates; κ is the vector of parameter estimates; ρ_i_ is the error term.

## 4. Results

### 4.1. Sample Description

The statistic characteristics of all participants are shown in Table 2. The average level of self-rated health for the total sample was 2.860 with a range from 1 to 5, which meant that the general health status of older women was between poor and fair. Aging women who participated in social activities accounted for 48.33% of the total sample. For the group who had social participations, the level of self-rated health status was 2.940, which was slightly higher than the average. For the group who had no social participations at all, the level of self-rated health status was 2.780, which was slightly lower than the average. The average age of all participants was 69.320 years old (SD = 7.220), and the average age of older women who participated in social activities was 68.610 years old (SD = 6.840). In total, 39% of aging women who had social participations took the responsibility of caring for grandchildren. The mean score of relationship with children for those who had social participation was 3.600, with a range from 1 to 5.

### 4.2. Impacts of Social Participation on Self-Rated Health of Aging Women

#### 4.2.1. Oprobit Regression Results

Table 3 reports the parameter estimates and marginal effects of the Oprobit model. It can be found that, regardless of whether control variables were considered, social participation always had a positive effect on the self-rated health of older women (*p* < 0.01). After the control variables were added, most variables still had significant impacts on the self-rated health of older women, whether it was a positive or negative impact. These results verified our Hypothesis 1 of this article. Additionally, as shown in the marginal effects in column (3) of Table 3, for those who participated in social activities, the probability to rate their health status as “fair” was increased by 1.1%, the probability to rate their health status as “good” was increased by 1.3%, and the probability to rate their health status as “very good” was increased by 1.8%. However, considering that some control variables may have potential endogenous problems, we do not discuss the results of these variables at this point.

#### 4.2.2. Impacts of Different Types of Social Participation on Self-Rated Health

We have confirmed H1, that social participation had a positive impact on self-rated health among aging women in China. However, amongst 10 different social activities provided in the CHARLS 2018 Questionnaire, we asked, what kind of social activities are more effective to improve the probability of a better health evaluation? Previous studies have categorized social activities based on different criteria [45,46,47]. Using content analysis of existing literature about social participation, Levasseur et al. concluded six proximal to distal levels of involvement of the individual based on the main goal of social activities [45]. In this study, we then categorized social activities into two types based on the taxonomy of social activities proposed by Levasseur et al. Type 1 were activities that did not directly contribute to others or the community, including (1) interacted with friends, (2) played mahjong, played chess, played cards, or went to a community club, (4) went to a sport, social, or other kind of club, (5) took part in a community-related organization, (8) attended an educational or training course, (9) stock investment, and (10) used the internet. Type 2 were activities that directly made contributions to others or the community, including (3) provided help to family, friends, or neighbors who do not live with you, (6) did voluntary or charity work, (7) cared for a sick or disabled adult who does not live with you.

Columns (2) and (3) in Table 4 report the results. Through comparison, we found that the probability of individuals who took part in Type 1 activities to rate their health as “very good” was increased by 1.7%, while the probability was increased by 2.8% in Type 2 activities. Type 2 activities that directly made contributions to others or the community had a more effective influence on better self-rated health. The results verified our Hypothesis 1a, that different types of social participation might have different effects on the probability of a better self-rated health among aging women in China.

#### 4.2.3. Heterogeneity Analysis

The previous have examined impacts of social participation on self-rated health of aging women and verified Hypothesis 1 and Hypothesis 1a. However, social participation is a personal behavior, and its impact on aging women’s health evaluation will also be influenced by other differences. The previous results were only average effects at the full sample level. In order to further verify and examine Hypothesis 1, we continued to investigate the heterogeneity of impacts of social participation on self-rated health of aging women, taking region, residence, and partners into consideration based on the Oprobit regression.

As shown in Table 5, except for the insignificant results of the subsample for the east region in column (2), results of other different subsamples were consistent with the full sample results; namely, social participation was significantly positively correlated with the self-rated health of aging women (*p* < 0.01).

From the perspective of regions, for aging women in the eastern region, social participation was not a significant factor affecting their health self-evaluation rate. However, for older women in the central and western regions, their participation in social activities significantly affected their health self-evaluation rate. Specifically, in the central region, the probability to rate the health status as “very good” was increased by 2.1%, which was higher than the average level of 1.8%; in the western region, the probability to rate the health status as “very good” was increased by 1.4%, which was lower than the average level.

From the perspective of residence, all different subsamples were consistent with the full sample results, namely, social participation was significantly positively correlated with self-rated health of aging women (*p* < 0.01). Specifically, older women participating in social activities in the urban-rural integration zone had the highest increase of 4.3% in the probability of self-evaluation as “very good”, while there was a 3.6% increase of the probability from urban areas, and the lowest increase was 1.1% from rural areas.

From the perspective of partners, older women who did not have a partner had a higher increase of probability to rate their health status as “very good”, compared with those who had a partner.

#### 4.2.4. The Mediating Effect of Caring for Grandchildren

In Table 6, Column (1) reports the estimated results of the Oprobit regression. As we already confirmed before, after adding control variables, social participation still had a significant impact on the self-rated health status of aging women (*p* < 0.01). Column (2) shows that the impacts of social participation on the responsibility of caring for grandchildren rejected the null hypothesis and verified our Hypothesis 2 (*p* < 0.01). Column (3) reports the results of Oprobit regression with the mediator and confirmed that the mediator functioned in the process. In summary, the test results strongly supported our Hypothesis 2; social participation positively influenced caring for grandchildren and caring for grandchildren had a positive impact on self-rated health amongst aging women in China.

### 4.3. Robustness Test

In order to verify the robustness of our results, this article adopted the following test methods:

#### 4.3.1. Narrow down the Targeted Sample

Considering the fact that the health self-evaluation is also influenced by personal physical condition [48,49], the result of self-rated health might be different among aging women who had a healthy physical condition and those who were disabled, even though they all participated in social activities. Therefore, in the robustness test, we modified the sample group and targeted those who were disabled. The results in column (2) of Table 7 show that the impacts of social participation on self-rated health results of the disabled group were still consistent and steady.

#### 4.3.2. Adjust Control Variables

In order to eliminate influences of individual characteristics and family characteristics on the outcome variable (self-rated health) and test the robustness of the Oprobit regression, we partially adjusted the control variables. While keeping insurance, income, and region, we added age group, bad habits, literacy, partner, pension, and disability (definitions and codes are shown in Table 8) as control variables and constructed a new Oprobit model. The results shown in column (3) of Table 7 were robust.

#### 4.3.3. Replacing the Measuring Question of the Outcome Variable

In this research, we used the question “Would you say your health is very good, good, fair, poor, or very poor?” from the CHARLS 2018 questionnaire to measure the outcome variable. As we explained in literature review, self-rated health is a subjective measurement and based on interviewees’ evaluations of their health [18,19,20,21]. Self-satisfaction of health is also the result of personal perception and reflects how they are satisfied with their health, and it can be regarded as a measurement to self-rated health. To conduct a robustness test, we replaced the original question with “How satisfied are you with your health?” from the CHARLS 2018 questionnaire. The results in column (4) of Table 7 show a significant positive relationship between social participation and the self-rated health level of aging women.

#### 4.3.4. Propensity Score Matching (PSM)

In order to correct the potential bias in the process of selecting variables, we used the PSM method to perform the test. Table 9 shows that, after robustness tests were performed using five methods: K-Nearest Neighbor (KNN), Radius Matching, Local Linear Regression, Mahalanobis Distance Matching (MDM), and Kernel Matching, the results of the average treatment effect for the treated (ATT) were consistent with the previous empirical results. To be specific, there was a significant positive relationship between social participation and self-rated health level of aging women in China. According to the results of the balancing test shown in Table 10, the standardization error of most variables after matching was less than 10%, which eliminated the imbalance among control variables and met the requirements of PSM. After the PSM test, we further verified the robustness of the results.

### 4.4. Endogenous Variable Processing

Due to the limitation of data and variables, empirical analysis may still have bias. In order to verify whether the positive impacts of social participation on older women were consistent and stable, we chose the relationship with children as our instrumental variable for the Hausman Test. After the test, it was shown that the *p*-value was smaller than 0.01. Therefore, relationships with children could be considered as an endogenous explanatory variable, which can be processed by the instrumental variable method. This article used two-stage least squares (2SLS) and limited information maximum likelihood (LIML) to conduct endogeneity testing. The results showed that the estimated coefficient value of LIML was very close to that of 2SLS. In addition, the influence coefficience of the instrumental variable on social participation in the first stage regression was statistically significant. The F-value was 19.87, which is greater than 10, and verified that there was no weak instrumental variable [50]. Through the second stage analysis in models (2) and (3) of Table 11, it can be found that, after the endogenous variable processing, social participation still significantly affected the health self-evaluation level of older women, which was consistent with our previous result.

## 5. Discussion

We used data from CHARLS 2018 and studied the relationship between social participation and self-rated health and considered the mediating effect of caring for grandchildren. The Oprobit regression confirmed that there was a positive association between social participation and the probability of a better self-rated health among aging women in China (Table 3). Different types of social activities proved to make different impacts on self-rated health (Table 4). Furthermore, as a mediator, the responsibility of caring for grandchildren was found to have a positive impact on self-rated health results (Table 6).

It is generally acknowledged that caring for grandchildren will make a great influence on grandmothers, whether it is to their daily life or to their health, even though women’s traditional roles as family caregivers may also contribute to their increased poverty and ill health in older age, according to the WHO [3] (p.20). Our results showed that aging women who took part in social activities had a higher possibility of caring for grandchildren (Table 2, Row 7) and led to a positive impact on self-rated health (Table 6, Column 3). Supporting these results, previous research has shown that taking care of grandchildren had a positive effect on both physical health and mental health of elderly people [51,52,53,54] and helped them to increase their self-esteem [55,56].

We categorized social activities based on whether it directly made contributions to others or the community and analyzed the impact of two types of activities on health (Table 4). The results shows that Type 2 activities, which directly made contributions to others or the community, had a more effective influence on better self-rated health. Many existing researches have supported this result: Oman et al. concluded that service to others and the altruistic features of volunteerism might impact the health of older people through cognitive and emotional pathways [57]; Krause et al. found that the elderly who provided assistance to others more often rated their health more favorably than those who were less involved in helping others [58]; Xia believed that the participation of the elderly in volunteer activities is an important way to achieve active aging and improves the subjective well-being and self-evaluated health of the elderly [59].

In the heterogeneity analysis, we found that, in the east region, social participation was not a significant factor affecting older women’s health self-evaluation rate, while in the central and west regions, social participation significantly affected their health self-evaluation rate (Table 5, Columns 2, 3, and 4). According to the active aging index designed by Liu and Yang, the development of active aging in China was unbalanced and the east region was the most developed area, while the western region was the least developed [60]. As many cities in eastern region, such as Shanghai, Hangzhou, and Guangzhou, have led the economic and administrative development all over the country, the eastern region has a relatively completed system of social support for older people. Under this condition, older people in the region start to pursue skill training and lifelong learning instead of amusement or exercises via social participation [60]. The self-satisfaction they get from social participation might be saturated, and social participation will no longer be a significant factor in self-rated health.

As for the residence, older women participating in social activities in urban–rural integration zones had the highest increase in the probability of self-evaluation as “very good”, compared to those from urban areas, and the lowest increase was from rural areas (Table 5, Columns 5, 6, and 7). Urban and rural differences have always been regarded as one of the major factors that affect the participation of the elderly in social activities, and this result might be explained by following studies: Social participation is influenced by physical and social aspects of the environment, such as urban design, traffic density and speed, infrastructures, and proximity to resources [61,62]; therefore, people in urban areas have better sources for social participation. Along with the impact of the one-child policy, the number of empty-nest families and single-living families in cities continues to increase. This situation has encouraged elderly people to participate in social activities [63]. Taking Japan as an example, in rural areas, the agricultural mode of production ensures tight social bonds [64], so elder people in rural areas do not heavily rely on social activities to achieve their social integration. As for why people in urban–rural integration zones had the highest increase in the probability of self-evaluation as “very good”, this could be regarded as the result of the integration process of urbanization. The government emphasized civil rights and granted them the same welfare treatment as the urban citizens, so they have improved their awareness of social participation and self-management [65]. However, there were few studies focused on elderly people in urban–rural integration zones, and we believe this is a field worthy of further study in the future.

Another consideration in the heterogeneity analysis is whether the aging women had partners. The results showed that older women who did not have a partner had a higher increase of the probability to rate their health status as “very good” after taking part in social activities, compared with those who had a partner (Table 5, Columns 8 and 9). Many studies supported the finding that widowed persons spent more time in social participation to seek social support [66,67], while married couples may not need additional forms of social support [68]. Given this, whether an older woman is a widowed or not, aging women without a partner have a higher possibility to socially participate, which increases the probability to rate their health status as “very good”.

To the best of our knowledge, this study is the first to target aging women in China and investigate the relationship between social participation and self-rated health with a mediator of caring for grandchildren. Our results verified the idea that social participation is an effective method to improve older women’ health, cope with the challenge of feminization in an aging population, and achieve the ultimate goal of active aging. However, this study still has some limitations. First, our research relied on the data from CHARLS 2018, which was collected three years ago, so might have a time lag on the results; Second, all variable choices were based on the CHALRS 2018 questionnaire, thus, we lacked sufficient variables, especially the scale of social participation, which was narrowed by only 10 social activities provided by the questionnaire. Additionally, the selection of instrumental variables was limited; Third, this study focused on aging women at a national average level; however, the family status and personal development of women are extremely unbalanced in China and the facts of social participation and health status vary within the country.

## 6. Conclusions

In the wave of global aging, the World Health Organization proposed the active aging policy framework, which applies to countries all over the world to deal with aging population challenges. As a national development strategy in China, active aging treats older people as essential human resources and encourages them to participate in social activities to achieve their own values in old age. Facing the fact that health problems have been more serious among older female groups, this study testified that social participation is beneficial to the self-rated health among aging women in China. Even though it is a compromise strategy for families to cope with the pressure of raising children, intergenerational care has been proved to have a positive effect on the self-rated health of older women. Therefore, Chinese society should recognize the value of intergenerational care for aging women and take a positive attitude towards getting old. In the future, more research should be done considering regional differences. In particular, the social participation for people in urban–rural integration zones is a new area worthy of further exploration. Consequently, more recommendations can be provided for our government to better face up to the demands of the elderly for social participation and strengthen policy supports to guarantee platforms and opportunities for the elderly to participate in social activities.

## Figures and Tables

**Figure 1 ijerph-18-05790-f001:**
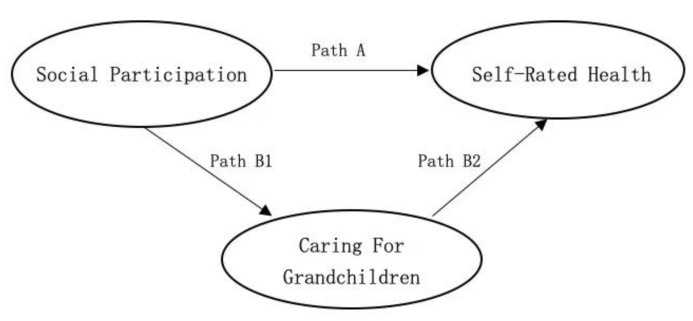
Conceptual framework.

**Table 1 ijerph-18-05790-t001:** Control variables.

Variable	Original Question	Codes
Age	What is your date of birth on your ID card or household register?	Calculated by 2018 minus the respondent’s birth year.
Residence	Was your address, BB000_W3, in the village or city/town?	1 = Central City/Town; 2 = Urban–Rural Integration Zone; 3 = Rural; 4 = Special Zone
Smoking	Have you ever chewed tobacco, smoked a pipe, smoked self-rolled cigarettes, or smoked cigarettes/cigars?	1 = Yes; 0 = No
Drinking	Did you drink any alcoholic beverages, such as beer, wine, or liquor in the past year? How often?	1 = Yes; 0 = No
Education	What is the highest level of education you have now (not including adult education)?	1 = No formal education (illiterate); 2 = Did not finish primary school; 3 = Middle school; 4 = High school and vocational school; 5 = college (associate degree) or above
Marriage	What is your marital status?	1 = Married with Spouse Present; 2 = Married But Not Living with Spouse Temporarily for Reasons Such as Work; 3 = Separated; 4 = Divorced; 5 = Widowed; 6 = Never Married
Minority	Are you Han or Ethnic Minorities?	1 = Han Nationality; 0 = Ethnic Minorities
Income	Did you receive any wage or bonus income in the past year?	1 = Yes; 0 = No
Insurance	Are you the policy holder/primary beneficiary of any of the types of health insurance listed below?	1 = Yes; 0 = No
Region	What is your address in [ZIWTime]?	Based on provided addresses, dividing the region into east, central, and west regions; 0 = East; 1 = Central; 2 = West.

**Table 2 ijerph-18-05790-t002:** Sample characteristics (*N* = 5015).

Variable	Total	Social Participation	Non-Social Participation	
	Mean (SD)	Mean (SD)	Mean (SD)	Range
Outcome Variable				
Self-rated Health	2.860 (1.000)	2.940 (0.980)	2.780 (1.010)	1–5
Explanatory Variable				
Social Participation	0.483 (0.500)	1.000 (0.000)	0.000 (0.000)	0–1
Mediator				
Caring for Grandchildren	0.360 (0.480)	0.390 (0.490)	0.330 (0.470)	0–1
Instrumental Variable				
Relationship with Children	3.578 (0.759)	3.600 (0.740)	3.550 (0.770)	1–5
Control Variable				
Age	69.320 (7.220)	68.610 (6.840)	69.960 (7.480)	60–108
Age2	4857.840 (1053.610)	4754.520 (988.890)	4950.090 (1098.570)	3600–11,664
Residence	2.520 (0.820)	2.430 (0.860)	2.620 (0.750)	1–4
Smoking	0.020 (0.130)	0.020 (0.140)	0.010 (0.110)	0–1
Drinking	0.130 (0.340)	0.150 (0.360)	0.110 (0.310)	0–1
Education	1.910 (0.990)	2.080 (1.040)	1.740 (0.910)	1–5
Marriage	2.180 (1.790)	2.190 (1.800)	2.170 (1.790)	1–6
Minority	0.920 (0.260)	0.930 (0.260)	0.920 (0.270)	0–1
Insurance	0.970 (0.180)	0.970 (0.170)	0.960 (0.200)	0–1
Income	0.070 (0.260)	0.070 (0.260)	0.070 (0.260)	0–1
Region	0.990 (0.820)	0.940 (0.820)	1.040 (0.820)	0–2

**Table 3 ijerph-18-05790-t003:** Impacts of social participation on self-rated health of aging women (*N* = 5015).

Variable	(1)	(2)	(3) Marginal Effects (dy/dx)		
	Oprobit	Oprobit	Fair	Good	Very Good
Social Participation	0.187 *** (0.030)	0.117 *** (0.032)	0.011 *** (0.003)	0.013 *** (0.004)	0.018 *** (0.005)
Age		−0.138 *** (0.041)	−0.013 *** (0.004)	−0.016 *** (0.005)	−0.021 *** (0.006)
Age2		0.001 *** (0)	0.000 *** (0)	0.000 *** (0)	0.000 *** (0)
Residence		−0.122 *** (0.021)	−0.011 *** (0.002)	−0.014 *** (0.002)	−0.018 *** (0.003)
Smoking		−0.0431 (0.132)	−0.004 (0.012)	−0.005 (0.015)	−0.007 (0.02)
Drinking		0.173 *** (0.048)	0.016 *** (0.005)	0.020 *** (0.005)	0.026 *** (0.007)
Education		0.0189 (0.018)	0.002 (0.002)	0.002 (0.002)	0.003 (0.003)
Marriage		−0.0314 *** (0.010)	−0.003 *** (0.001)	−0.004 *** (0.001)	−0.005 *** (0.001)
Minority		−0.00991 (0.062)	−0.001 (0.006)	−0.001 (0.007)	−0.001 (0.009)
Insurance		−0.0498 (0.089)	−0.005 (0.008)	−0.006 (0.01)	−0.008 (0.013)
Income		0.304 *** (0.060)	0.028 *** (0.006)	0.034 *** (0.007)	0.046 *** (0.009)
Central Region		−0.178 *** (0.039)	−0.014 *** (0.003)	−0.020 *** (0.005)	−0.028 *** (0.006)
West Region		−0.210 *** (0.040)	−0.018 *** (0.004)	−0.024 *** (0.005)	−0.033 *** (0.006)
*N*	5015	4575	4575	4575	4575
pseudo R2	0.003	0.016			

Note: Standard errors in parentheses; *** *p* < 0.01; dy/dx for factor levels is the discrete change from the base level; column (1) and column (2) reported the parameter regression coefficient value; column (3) reported marginal effects (%).

**Table 4 ijerph-18-05790-t004:** Different types of social participation and self-rated health.

Variable	Oprobit (dy/dx)		
	(1)	(2)	(3)
	Self-Rated Health (Very Good)	Self-Rated Health (Very Good)	Self-Rated Health (Very Good)
Social Participation	0.018 *** (0.005)		
Type 1 Activity		0.017 *** (0.005)	
Type 2 Activity			0.028 *** (0.007)
Control Variables	Yes	Yes	Yes
*N*	4575	4575	4575
pseudo R2	0.016	0.016	0.016

Note: standard errors in parentheses; *** *p* < 0.01; dy/dx for factor levels is the discrete change from the base level; Column (1) to Column (4) report marginal effects (%).

**Table 5 ijerph-18-05790-t005:** Heterogeneity analysis (dy/dx) (*N* = 5015).

Variable	Pooled	Subsample: Region			Subsample: Residence			Subsample: Partner	
	(1) Full Sample	(2) East	(3) Central	(4) West	(5) Urban	(6) Urban–Rural Integration Zone	(7) Rural	(8) Yes	(9) No
	Very Good	Very Good	Very Good	Very Good	Very Good	Very Good	Very Good	Very Good	Very Good
Social Participation	0.018 *** (0.005)	0.018 (0.011)	0.021 *** (0.008)	0.014 ** (0.006)	0.036 *** (0.012)	0.043 ** (0.019)	0.011 ** (0.006)	0.012 * (0.006)	0.030 *** (0.008)
Region	Yes	No	No	No	Yes	Yes	Yes	Yes	Yes
Residence	Yes	Yes	Yes	Yes	No	No	No	Yes	Yes
Partner	Yes	Yes	Yes	Yes	Yes	Yes	Yes	No	No
Others	Yes	Yes	Yes	Yes	Yes	Yes	Yes	Yes	Yes
*N*	4575	1547	1471	1557	954	292	3315	3362	1213
pseudo R2	0.0156	0.0151	0.0177	0.0130	0.0176	0.0234	0.0127	0.0134	0.0207

Note: Standard errors in parentheses; * *p* < 0.1, ** *p* < 0.05, *** *p* < 0.01; dy/dx for factor levels is the discrete change from the base level; Column (1) to Column (9) reported marginal effects (%).

**Table 6 ijerph-18-05790-t006:** The mediating effect of caring for grandchildren (dy/dx).

Variable	Oprobit	Probit	Oprobit
	(1)	(2)	(3)
	Self-Rated Health (Very Good)	Caring for Grandchildren (Yes)	Self-Rated Health (Very Good)
Social Participation	0.018 *** (0.005)	0.029 * (0.017)	0.016 *** (0.006)
Caring for Grandchildren			0.018 *** (0.006)
Control Variables	Yes	Yes	Yes
*N*	4575	3006	2784
pseudo R2	0.016	0.124	0.018

Note: Standard errors in parentheses; * *p* < 0.1, *** *p* < 0.01; dy/dx for factor levels is the discrete change from the base level; Column (1) to (3) report marginal effects (%).

**Table 7 ijerph-18-05790-t007:** Robustness test.

Variable	Oprobit			
	(1)	(2)	(3)	(4)
	Self-Rated Health	Self-Rated Health	Self-Rated Health	Health Satisfaction
Social Participation	0.117 *** (0.032)	0.212 ** (0.084)	0.100 ** (0.046)	0.082 ** (0.032)
Age	−0.138 *** (0.041)	−0.146 (0.099)		−0.096 ** (0.042)
Age2	0.001 *** (0.000)	0.001 (0.001)		0.001 ** (0.000)
Residence	−0.122 *** (0.021)	−0.083 (0.059)		−0.057 *** (0.021)
Smoking	−0.043 (0.132)	0.030 (0.359)		0.019 (0.132)
Drinking	0.173 *** (0.048)	0.097 (0.127)		0.116 ** (0.047)
Education	0.019 (0.018)	0.060 (0.050)		−0.081 *** (0.018)
Marriage	−0.031 *** (0.010)	−0.057 ** (0.025)		−0.004 (0.010)
Minority	−0.010 (0.062)	0.219 (0.157)		−0.035 (0.062)
Insurance	−0.050 (0.089)	−0.028 (0.228)	−0.013 (0.120)	−0.104 (0.089)
Income	0.304 *** (0.060)	0.273 (0.188)	0.260 *** (0.086)	0.277 *** (0.060)
Central Region	−0.178 *** (0.039)	0.056 (0.106)	−0.158 *** (0.057)	−0.021 (0.039)
West Region	−0.210 *** (0.040)	−0.081 (0.105)	−0.195 *** (0.056)	−0.063 (0.039)
Age Gap			0.105 * (0.061)	
Bad Habit			0.144 ** (0.064)	
Literature			0.101 * (0.054)	
Partner			0.153 *** (0.055)	
Pension			−0.008 (0.066)	
Disability			−0.606 *** (0.055)	
*N*	4575	686	2182	4525
Pseudo R2	0.0156	0.0167	0.0321	0.0054

Note: standard errors in parentheses; * *p* < 0.1, ** *p* < 0.05, *** *p* < 0.01; Column (1) to Column (4) show the parameter regression coefficient values.

**Table 8 ijerph-18-05790-t008:** Added control variables for the robustness test.

Variable	Definition	Codes
Age Group	Categorized participants into 1. Youth; 2. Middle-Age; 3. Old Age; 4. Elderly; 5. Senior	1 = 1–44 years old; 2 = 45–59 years old; 3 = 60–74 years old; 4 = 75–89 years old; 5 =≥ 90 years old
Bad Habit	Do you smoke or drink?	1 = yes; 0 = no
Literature	Are you literate?	1 = yes; 0 = no
Partner	Do you have a partner?	1 = yes; 0 = no
Pension	Do you have a pension?	1 = yes; 0 = no
Disability	Are you disabled?	1 = yes; 0 = no

**Table 9 ijerph-18-05790-t009:** Propensity score matching results.

Methods	Sample	Have Social Participation = (1)	No Social Participation = (2)	ATT = (1)–(2)	SE	*t*-Value
KNN (K = 4)	Unmatched	2.936	2.792	0.144	0.029	4.93 ***
	Matched	2.936	2.825	0.111	0.035	3.17 ***
Radius Matching	Unmatched	2.936	2.792	0.144	0.029	4.93 ***
	Matched	2.936	2.832	0.104	0.040	2.62 ***
Local Linear Regression	Unmatched	2.936	2.792	0.144	0.029	4.93 ***
	Matched	2.936	2.827	0.109	0.040	2.75 ***
MDM	Unmatched	2.936	2.792	0.144	0.029	4.93 ***
	Matched	2.936	2.858	0.078	0.034	2.27 ***
Kernel Matching	Unmatched	2.936	2.792	0.144	0.029	4.93 ***
	Matched	2.936	2.824	0.112	0.031	3.61 ***

Note: *** *p* < 0.01.

**Table 10 ijerph-18-05790-t010:** Balancing test.

Methods	Sample	Ps R^2^	LR chi^2^	*p* > chi^2^	Mean Bias	Med Bias
KNN (K = 4)	Unmatched	0.035	224.97	0.000	11.2	10.1
	Matched	0.002	10.35	0.585	2.8	3.3
Radius Matching	Unmatched	0.035	224.97	0.000	11.2	10.1
	Matched	0.002	14.15	0.291	2.2	1.4
Local Linear Regression	Unmatched	0.035	224.97	0.000	11.2	10.1
	Matched	0.002	14.70	0.258	2.2	1.4
MDM	Unmatched	0.035	224.97	0.000	11.2	10.1
	Matched	0.003	16.34	0.176	2.0	0.7
Kernel Matching	Unmatched	0.035	224.97	0.000	11.2	10.1
	Matched	0.001	3.99	0.984	1.4	1.3

**Table 11 ijerph-18-05790-t011:** Instrumental variable estimation.

Variable	(1)	(2)		(3)	
	Oprobit (dy/dx)	2SLS		LIML	
		First Stage	Second Stage	First Stage	Second Stage
Social Participation	1.775 *** (0.005)		11.75 * (6.937)		11.751 * (6.937)
Instrumental Variable		0.016 * (0.010)		0.016 * (0.010)	
Control Variables	Yes	Yes	Yes	Yes	Yes
*N*	4575	4510	4510	4510	4510
F		19.87		19.87	
pseudo R2 or R2	0.016	0.0482		0.0482	

Note: standard errors in parentheses; * *p* < 0.1, *** *p* < 0.01; dy/dx for factor levels is the discrete change from the base level; Column (1) shows marginal effects (%); Column(2) and (3) show parameter regression coefficient values.

## Data Availability

Datasets are distributable only by the CHARLS team. They are available in the public domain on the CHARLS website, http://charls.pku.edu.cn/zh-CN (accessed on 10 April 2021), and are also available on request from the corresponding author.

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
