# Peer review of "Impacts of Social Participation on Self-Rated Health of Aging Women in China: With a Mediating Role of Caring for Grandchildren"

_ijerph, 2021, doi:10.3390/ijerph18115790_

Round 1

Reviewer 1 Report

DEFICIENCIES:

The study presents an interesting and yet not well-covered field in literature. It supports the need to investigate more in this field. Nevertheless, the present article has to be reviewed in each section. It also needs a review of language/grammar.

INTRODUCTION:

Lines 47-59: This part represents a comment on the WHO Report. Too much 'space' is devoted to this comment. Therefore, I suggest moving this part below to the literature review section. Try to add a better comparison to other studies and / or reports, proceedings to this section, and don't use just one report. The exclusive use of the Report is not a benefit to the study.

At the end of the Introduction, add a small section in order to show what you are presenting below. What do you include in the next paragraphs? Please, provide a brief explanation.

LITERATURE REVIEW:

In addition, many scholars have found that the 87 active participation of the elderly in voluntary service is beneficial to enhance happiness 88 and realize the value of life [15-17]”. (Lines 87-88). You can skyrocket this concept by improving your literature review. What you cite open literally "a world". So, a better and improved clarification of what these authors (15-17) assert, could give a surplus to the entire dissertation. In addition, try to add other studies that face the active participation of elderly people. In this way, you will present a concept as a whole.

Lines 99-100: Why do you not try to compare the family structure in other countries as well? For example, in the European zone, is it the same? What is the attitude in other parts of the world? This study, being of a scientific nature, should make a comparison with other studies carried out elsewhere in the world, not focus solely on the Chinese aspect. For this reason, I advise you to make some interesting comparisons.

METHOD:

The whole structure of the methodology (steps carried out to perform the analysis) should be simplified. To do this, I suggest inserting a table, writing the order of the inclusion criteria, and, possibly, adding also what exclusion criteria have been considered.

Lines 147-148: What are the others inclusion and exclusion criteria to select your sample? Did you use only the age? It is too little exhaustive to use a single criterion for the selection of the sample. Add any other criteria you used or, if you haven't used any others, please, add them now to your analysis.

Lines 152-154: You should simplify these sentences. There are 3 sentences that said the same things. You can explain this with only one sentence.

RESULTS:

Line 253: In table 2, SD should not be shown in another column, but it is preferable to see it among brackets. E.g.: mean(SD) -> 2860(1) etc ...

I believe that the Authors did a good job, accurately presenting the results. They are easy to understand and the work they have done is appreciable.

DISCUSSION:

Lines 426-427: Could you comment on the results of the studies cited in order to give the possibility to the reader to understand why you choose them to compare with your results?

CONCLUSIONS:

In general, conclusions need a small major effort. They need to be correlated with data and link to the objectives of the present study. Future research directions are not presented, thus, add these important aspects.

What is the gap that the study fills?

What advanced knowledge is included in this study?

Author Response

Dear Reviewer,

We appreciate the time and effort that you dedicated to providing feedback on our manuscript and are grateful for the insightful comments on and valuable improvements to our paper.  Please see the attachment for our responses. 

Thanks again!

Sincerely,

Xiangling Wu

Reviewer 2 Report

This research is timely for all nations and not just China, hence politically it is important to publish it. However, selecting caring for Grandchildren as a mediator when the actual question used states "During last year, did you/ your spouse spend time in taking care of your grandchildren?" Once in 12 months or 365 days is very limited and possibly calls into question the analyses- either that or once in 12 months of caring for Grandchildren has a very large impact. If the question had been at the same time frame as the social activities ie once per month that would have been better. It is recognised that the authors might not have access to data showing whether the participants cared for their grand children more frequently than once a year. Likewise information on population demographics in the East, Central and West Regions would be helpful as it is my perception that many of the cities in the developed East have a predominance of young people in their population and the more rural parts in the West of China consist mostly of more elderly people. Perhaps my perception is wrong, but some figures to provide the reader with more information on this would be helpful as this information might help with interpretation of results. 

Author Response

Dear Reviewer,

We appreciate the time and effort that you dedicated to providing feedback on our manuscript and are grateful for the insightful comments on and valuable improvements to our paper.

Also, thanks for your understanding that we have limited choices in the selection of questions for the mediator. As for the population distribution in different regions, the data presented that the amount of participants in east, central and west regions is very close to each other, so we didn't emphasize in sample description, but we have enriched our explanation in discussion  part regarding to your suggestion.

Thanks you so much for your help!

Best,

Xiangling Wu